# HIGH-DIMENSIONAL GEOMETRIC STREAMING FOR NEARLY LOW RANK DATA

## ABSTRACT

We study streaming algorithms for the outer $(d-k)$-radius estimation of a set of points $a_1, \ldots, a_n \in \mathbb{R}^d$. The problem asks to compute the minimum over all $k$-dimensional flats $F$ of $\max_i d(a_i, F)$, where $d(u, F)$ denotes the distance of a point $u$ from the flat $F$. This problem has been extensively studied in earlier works (Varadarajan et al., SIAM J. Comput. 2006) over a wide range of values of $d$, $k$ and $d - k$. The earlier algorithms are based on SDP relaxations of the problem and are not applicable in the streaming setting where we do not have space to store all the rows that we see. We give an efficient streaming coreset algorithm that selects $\mathrm{poly}(k, \log n)$ rows and at the end outputs a $\mathrm{poly}(k, \log n)$ approximation to the outer $(d-k)$-radius. The algorithm only uses $d \cdot \mathrm{poly}(k, \log n)$ bits of space and runs in an overall time of $O(\mathrm{nnz}(A) \cdot \log n + \mathrm{poly}(d, \log n))$, where $\mathrm{nnz}(A)$ denotes the number of nonzero entries in the $n \times d$ matrix $A$ with rows given by $a_1, \ldots, a_n \in \mathbb{R}^d$.

In a recent work, Woodruff and Yasuda (FOCS 2022), give streaming algorithms for high-dimensional geometric problems such as width estimation, convex hull estimation, volume estimation etc. Their algorithms require $\Omega(d^2)$ bits of space and have an $\Omega(\sqrt{d})$ multiplicative approximation factor even when the rows $a_1, \ldots, a_n$ are "almost" spanned by a $k$ dimensional subspace. We show that when the rows are $a_1, \ldots, a_n$ are "almost" spanned by a $k$ dimensional space, our streaming coreset construction algorithm can be used to obtain algorithms that use only $O(d \cdot \mathrm{poly}(k, \log n))$ bits of space and have a multiplicative error of $O(\mathrm{poly}(k, \log n))$. When $k \ll d$, our algorithms use a much smaller amount of space while guaranteeing a better approximation. We pay an additive error depending on how close the rows $a_1, \ldots, a_n$ to being spanned by a rank $k$ subspace.

As another application of our algorithm, we show that our streaming coreset can also be used to obtain approximations to the $\ell_p$ subspace approximation problem using exponential random variables to embed the $\ell_p$ subspace approximation problem into an instance of the $\ell_\infty$ subspace approximation problem.

## 1 INTRODUCTION

Modern datasets are usually very high dimensional and have a large number of data points. Storing the entire dataset to analyze them is often impractical and in certain settings impossible. In recent years, streaming algorithms have emerged as a way to process and understand the datasets in both a space and time efficient manner. In a single-pass streaming setting, the algorithm is allowed to make only a single pass over the entire dataset and is required to output a "summary" of the dataset that is useful to solve a certain problem. In this work, we focus on streaming algorithms for geometric problems such as subspace approximation, width estimation, etc. Suppose that we are given a set of $d$-dimensional points $a_1, \ldots, a_n$ and a dimension parameter $k$. Given a subspace $V$, we define $d(a, V)$ to be distance between the point $a$ and subspace $V$ given by $\min_{v \in V} \|a - v\|_2$. The $\ell_p$ subspace approximation problem (Deshpande et al., 2011b), for $p \in [1, \infty]$, asks to find a $k$-dimensional subspace that minimizes $(\sum_{i=1}^n d(a_i, V)^p)^{1/p}$.

Note that for $p = \infty$, we want to find a $k$-dimensional subspace that minimizes the maximum distance from the given set of points. Related to the $\ell_\infty$ subspace approximation problem is the widely studied outer $(d-k)$ radius estimation problem (Varadarajan et al., 2007) which instead asks for a

$k$-dimensional flat[1] $F$ that minimizes $\max_{i \in [n]} d(a_i, F)$. The outer $(d - k)$ radius is a measure of how far the point set is from being a $k$-dimensional flat. Varadarajan et al. (2007) give a polynomial time algorithm for approximating the outer $(d - k)$ radius up to a $O(\sqrt{\log n})$ multiplicative factor. Their algorithm is based on rounding of a semidefinite program (SDP) relaxation. When $n$ and $d$ are very large, their algorithm is not practical and cannot be implemented in the streaming setting. We give a time and space efficient single pass streaming algorithm that approximates the outer $(d - k)$ radius up to a $\text{poly}(k, \log n)$ factor. Typically, the value of $k$ used is much smaller than $n$ and $d$ since in many settings, we have that the $n \times d$ matrix $A$ is a noisy version of the underlying rank $k$ matrix.

Our algorithm is based on constructing a *strong coreset* for approximating $\max_i d(a_i, V)$ for any $k$-dimensional subspace $V$. When run on the stream of points $a_1, \ldots, a_n$, our algorithm selects a subset $S \subseteq [n]$ of points, $|S| = \text{poly}(k, \log n)$ such that for all $k$ dimensional subspace $V$, $\max_{i \in S} d(a_i, V) \leq \max_{i \in [n]} d(a_i, V) \leq \text{poly}(k, \log n) \max_{i \in S} d(a_i, V)$ which implies that the optimal solution to the $\ell_\infty$ subspace approximation problem on the point set $(a_i)_{i \in S}$ is a $\text{poly}(k, \log n)$ approximation to the $\ell_\infty$ subspace approximation problem on the point set $(a_i)_{i \in [n]}$. We prove:

**Theorem 1.1** (Informal). *Given a parameter $k$ and $n$ points $a_1, \ldots, a_n \in \mathbb{R}^d$, all with integer coordinates bounded in absolute value by $\text{poly}(n)$, there is a* deterministic *single-pass streaming algorithm that selects a subset $S \subseteq [n]$ of points, $|S| = O(k^3 \log^2 n)$ such that for all $k$-dimensional subspaces $V$,*

$$\max_{i \in S} d(a_i, V) \leq \max_{i \in [n]} d(a_i, V) \leq O(k^{3/2} \log n) \max_{i \in S} d(a_i, V).$$

*The streaming algorithm uses only $O(d \cdot k^3 \log^3 n)$ bits of space and can be implemented in time $O(\text{nnz}(A) \log n + d \, \text{poly}(k, \log n))$.*

In this result and its applications, the size of the set $|S|$ can be replaced with $O(k \log^2 \kappa)$ where $\kappa$ is a suitably defined rank-$k$ condition number. For many practical datasets, $\kappa$ is small and we accordingly incur a distortion of only $O(\sqrt{k} \log \kappa)$ instead of $O(k^{3/2} \log n)$ in all our results.

Using a simple proof, we then show that if we run the coreset construction algorithm on the point set $0 = a_1 - a_1, a_2 - a_1, \ldots, a_n - a_1$ and if the algorithm selects a subset $S \subseteq [n]$, then the $\ell_\infty$ subspace approximation cost of the point set $(a_i - a_1)_{i \in S}$ is a $\text{poly}(k, \log n)$ approximation to the outer $(d - k)$ radius of the points $(a_i)_{i \in [n]}$. As the coreset is small, we can use any existing algorithm to approximate the $\ell_\infty$ subspace approximation cost of the points in the coreset. We show that as the coreset only has $\text{poly}(k, \log n)$ points, the top rank-$k$ singular subspace is a $\text{poly}(k, \log n)$ approximation to the $\ell_\infty$ subspace approximation problem. Hence, we can obtain a $\text{poly}(k, \log n)$ approximation to the outer $(d - k)$ radius without using *any* SDP relaxations.

We then turn to the $\ell_p$ subspace approximation for general $p \in [1, \infty)$. For $p = 2$, the $\ell_2$ subspace approximation problem is equivalent to the Frobenius norm low rank approximation problem. There are a wide variety of algorithms for this problem in the offline and streaming settings. For example, the FrequentDirections sketch Ghashami et al. (2016) is a deterministic algorithm that uses only $O(\varepsilon^{-1} dk \log n)$ bits of space and outputs a $1 + \varepsilon$ approximation to the $\ell_2$ subspace approximation problem. A downside of the FrequentDirections sketch is that it is *not* a weighted subset of the input rows and hence is not preferred in some cases as it is harder to interpret the sketch, does not preserve sparsity, etc. Braverman et al. (2020) give a randomized algorithm that samples and scales $\text{poly}(k, \log n)$ rows in the stream and at the end outputs a Projection-Cost Preserving (PCP) sketch (Cohen et al., 2015; Feldman et al., 2020). The algorithm uses $\varepsilon^{-2} d \, \text{poly}(k, \log n)$ bits of space and the sketch can be used to obtain a $1 + \varepsilon$ approximation to the $\ell_2$-Subspace Approximation problem. In addition, their procedure works even in the *online* setting where a selected row is never discarded.

For $p \notin \{2, \infty\}$, much less is known in the streaming setting. In the offline setting, Deshpande and Varadarajan (2007) gave a sampling based algorithm for all $p \geq 1$ that outputs a bicriteria solution for the $\ell_p$ subspace approximation problem. Later, in Deshpande et al. (2011a), they give a polynomial time $O(\sqrt{p})$ factor approximation algorithm for the $\ell_p$ subspace approximation problem for all $p \geq 2$. Assuming the Unique Games Conjecture, they show that it is hard to approximate the cost to a smaller than $O(\sqrt{p})$ factor. For $1 \leq p \leq 2$, Clarkson and Woodruff (2015) gave an input sparsity time algorithm that computes a $1 + \varepsilon$ approximation but they have an $\exp(\text{poly}(k/\varepsilon))$ term in their

---

[1]A $k$ dimensional flat is defined as a $k$ dimensional subspace that is translated by some $c$.

running time. The $O(\sqrt{p})$ factor approximation algorithm of Deshpande et al. (2011a) is based on convex relaxations is not applicable in the streaming setting of this paper. In a recent work, Deshpande and Pratap (2023) observed the lack of streaming algorithms for $\ell_p$ subspace approximation that also have the subset selection property that our coresets have. They give a subset selection algorithm for the $\ell_p$ subspace approximation problem but their results have a weaker additive error guarantee. They leave open the subset selection algorithms that give a multiplicative approximation to the $\ell_p$ subspace approximation problem. In a recent work, Woodruff and Yasuda (2023) answered the question of Deshpande and Pratap (2023) in the affirmative by giving a subset selection algorithm the computes a strong coreset with $O((k/\varepsilon)^{O(p)} \operatorname{polylog}(n))$ rows that can approximate the cost of any $k$-dimensional space up to a $1 \pm \varepsilon$ factor. Selecting $k^{O(p)}$ rows is prohibitive when $p$ is large.

Towards resolving this issue, we show that the coreset construction algorithm from Theorem 1.1 can be used to obtain a streaming algorithm that selects a subset of (weighted) points which can be used to obtain multiplicative approximations to the $\ell_p$ subspace approximation problem. We use the *min-stability* property of exponential random variables to embed an instance of $\ell_p$ subspace approximation to an instance of $\ell_\infty$ subspace approximation. Then, we can directly use the coreset for $\ell_\infty$ subspace approximation to obtain a solution that has a multiplicative guarantee. The technique of using exponential random variables to embed $\ell_p$ problems to $\ell_\infty$ problems has been employed in previous works, e.g., Cohen et al. (2014); Woodruff and Zhang (2013). While the algorithm (scale by exponential random variables and run the coreset for $\ell_\infty$ subspace approximation) turns out to be fairly simple, the analysis that the algorithm works is involved and the net argument we apply crucially uses the fact that the coreset is small to obtain good guarantees on the approximation factor. We are not aware of any previous works exploring the use of exponential random variables in the context of the $\ell_p$ subspace approximation problem. We obtain the following result:

**Theorem 1.2** (Informal). *Given $p \geq 1$, a dimension parameter $k$, and $n$ points $a_1, \ldots, a_n \in \mathbb{R}^d$ with integer coordinates bounded in absolute value by $\operatorname{poly}(n)$, there is a randomized streaming algorithm that selects a subset $S \subseteq [n]$, $|S| = O(k^3 \log^2 n)$ and assigns a weight $w_i \geq 0$ for $i \in S$ such that if*

$$\tilde{V} = \arg\min_{k\text{-dim } V} \max_{i \in S} w_i \cdot d(a_i, V),$$

*then $(\sum_{i=1}^n d(a_i, \tilde{V})^p)^{1/p} \leq k^{3/2+5/p} \operatorname{poly}(\log n) \min_{k\text{-dim } V}(\sum_{i=1}^n d(a_i, V)^p)^{1/p}$. The algorithm only uses $O(d \cdot k^3 \log^3 n)$ bits of space and runs in $O(\operatorname{nnz}(A) \log n + d \operatorname{poly}(k, \log n))$ time.*

We note that we give a *weak coreset* that can be used to compute an approximate solution whereas Woodruff and Yasuda (2023) give a strong coreset which can be used to approximate the cost of any $k$ dimensional subspace.

We then show that recent algorithms of Woodruff and Yasuda (2022) can be improved using our coreset construction algorithm when the data points $a_1, \ldots, a_n$ are "approximately" spanned by a low rank subspace. They give streaming algorithms for a host of geometric problems such as width estimation, volume estimation, Löwner-John ellipsoid, etc. The main ingredient of their algorithms is a deterministic $\ell_\infty$ "subspace embedding": their algorithm streams through rows of an $n \times d$ matrix $A$ and selects a subset of rows $S \subseteq [n]$, $|S| = O(d \log n)$ with the property that for all $x$,

$$\|A_S x\|_\infty \leq \|Ax\|_\infty \leq \sqrt{d \log n} \|A_S x\|_\infty.$$

Here $\|x\|_\infty \doteq \max_i |x_i|$ and $A_S$ is the matrix $A$ restricted to only those rows in $S$. When the matrix $A$ has rank $d$, their algorithm necessarily needs $\Omega(d^2)$ bits of space which is prohibitive when $d$ is very large. In practice, many datasets are very well represented by a matrix with far lower rank than $d$. Since that algorithm of Woodruff and Yasuda (2022) aims for a strong subspace embedding guarantee, their coreset construction algorithm selects $\Omega(d)$ rows even when the matrix can be approximated very well by a low rank matrix. We show that if $S$ is the coreset constructed by the algorithm in Theorem 1.1, then for all *unit vectors* $x$, $\|A_S x\|_\infty \leq \|Ax\|_\infty \leq Ck^{3/2} \log n \|A_S x\|_\infty + (Ck \log n)\Delta$, where $\Delta$ denotes the optimal rank-$k$ $\ell_\infty$ subspace approximation cost of the matrix $A$. When *all* the rows of the matrix $A$ are close to a rank $k$ subspace, then $\Delta$ is small and the above guarantee directly leads to improvements over their algorithm both in terms of space complexity and approximation ratios.

We include experiments to show that our algorithm works well in practice and that the coreset computed by the algorithm gives accurate answers to various queries. We emphasize that our algorithm is very efficient and can be implemented to run quickly on huge datasets (see Remark 3.4).

**Relevance to Machine Learning.** Our work continues the long line of work in the area of subspace approximation and low rank approximation with different error metrics that has been of interest in the Machine Learning community. Previous works study problems such as $\ell_1$ subspace approximation (Hardt and Moitra, 2013), entry wise $\ell_p$ low rank approximation (Chierichetti et al., 2017; Dan et al., 2019), Column subset selection for entrywise $\ell_p$ norm and other error metrics (Song et al., 2019).

Our algorithms for geometric streaming problems such as convex hull estimation have applications for robust classification (Provost and Fawcett, 2001; Fawcett and Niculescu-Mizil, 2007).

## 2 PRELIMINARIES

For integer $n \geq 1$, we use $[n]$ to denotes the set $\{1, \ldots, n\}$. For an $n \times d$ matrix $A$, we use $a_i \in \mathbb{R}^d$ to denote the $i$-th row. If $S \subseteq [n]$, then $A_S$ denotes the submatrix formed by the rows in the set $S$. For $x \in \mathbb{R}^d$ and $p \geq 1$, $\|x\|_p$ denotes the $\ell_p$ norm of $x$ defined as $(\sum_{i=1}^d |x_i|^p)^{1/p}$ and $\|x\|_\infty \doteq \max_i |x_i|$. Given a matrix $A$, $\|A\|_F$ to denote the Frobenius norm and $\|A\|_{p,2}$ to denote the $\ell_p$ norm of the $n$-dimensional vector $(\|a_1\|_2, \ldots, \|a_n\|_2)$. Given a matrix $A$, we use $[A]_k$ to denote the best rank-$k$ approximation of $A$ in Frobenius norm. This can be obtained by truncating the singular value decomposition of $A$ to the top $k$ singular values.

For an arbitrary $k$ dimensional subspace $V \in \mathbb{R}^d$, we use $\mathbb{P}_V$ to denote the orthogonal projection matrix onto the subspace $V$, i.e., for any $x \in \mathbb{R}^d$, $\mathbb{P}_V \cdot x$ is the closest (in Euclidean norm) vector to $x$ in $V$. So, $d(x, V) = \|(I - \mathbb{P}_V)x\|_2$ and $\|A(I - \mathbb{P}_V)\|_{\infty,2} = \max_i \|(I - \mathbb{P}_V)a_i\|_2 = \max_i d(a_i, V)$.

## 3 $\ell_\infty$ LOW RANK APPROXIMATION AND OUTER RADIUS

As discussed in the introduction, we want to compute a *strong coreset*, i.e., a subset $S$ of rows of $A$ such that for all $k$-dimensional subspaces $V$,

$$\|A_S(I - \mathbb{P}_V)\|_{\infty,2} \leq \|A(I - \mathbb{P}_V)\|_{\infty,2} \leq f \cdot \|A_S(I - \mathbb{P}_V)\|_{\infty,2}$$

for a small factor $f$. Consider the following simple algorithm: we initiate $S \leftarrow \emptyset$ and stream through the rows $a_1, \ldots, a_n$. When processing rows $a_i$, if there exists a rank-$k$ projection matrix $P$ such that $\|a_i^\top(I - P)\|_2 > \|A_S(I - P)\|_F$, then we update $S \leftarrow S \cup \{i\}$. Otherwise, we proceed to the next row. Now consider the set $S$ at the end of the stream and let $V$ be an arbitrary $k$ dimensional subspace. Let $i$ be such that $\|A(I - \mathbb{P}_V)\|_{\infty,2} = \|a_i^\top(I - \mathbb{P}_V)\|_2$. If $i \in S$, then we already have $\|A_S(I - \mathbb{P}_V)\|_{\infty,2} = \|A(I - \mathbb{P}_V)\|_{\infty,2}$. If $i \notin S$, then by the algorithm, we have that $\|a_i^\top(I - \mathbb{P}_V)\|_2 \leq \|A_{S<i}(I - \mathbb{P}_V)\|_F$. Now, $\|A_{S<i}(I - \mathbb{P}_V)\|_F^2 \leq |S| \|A_S(I - \mathbb{P}_V)\|_{\infty,2}^2$. Hence, $\|A(I - \mathbb{P}_V)\|_{\infty,2} = \|a_i^\top(I - \mathbb{P}_V)\|_2 \leq \sqrt{|S|} \|A_S(I - \mathbb{P}_V)\|_{\infty,2}$. So, we have for all rank-$k$ subspaces $V$ that $\|A_S(I - \mathbb{P}_V)\|_{\infty,2} \leq \|A(I - \mathbb{P}_V)\|_{\infty,2} \leq \sqrt{|S|} \|A_S(I - \mathbb{P}_V)\|_{\infty,2}$. Now, if we can show that $S$ can not be too large, we obtain that $A_S$ is a strong coreset with a small distortion.

To show that $S$ is not too large, we appeal to rank-$k$ *online* ridge leverage scores. In the offline setting, ridge leverage scores have been employed by Cohen et al. (2017) as a suitable modification of the usual $\ell_2$-leverage scores to obtain fast algorithms for $\ell_2$ low rank approximation. Later, Braverman et al. (2020) defined online ridge leverage scores and showed that they can be used to compute low rank approximations in the *online* model. They also showed that for well-conditioned instances, the sum of the online ridge leverage scores is small. Our main observation is that for the set $S$ constructed as described, the online ridge leverage score of *every* row in $A_S$ is large. As the sum of online ridge leverage scores is not large, we obtain that there cannot be too many rows in $A_S$.

One issue we have to solve to implement this algorithm is given $a_i$ and $A_S$, how can we efficiently know if there exists a rank-$k$ subspace $V$ such that $\|a_i(I - \mathbb{P}_V)\|_2 > \|A_S(I - \mathbb{P}_V)\|_F$? As we argued above, for all the rows in $A_S$, the online ridge leverage score is large. So, we instead modify the algorithm to add a row $a_i$ if its online ridge leverage score with respect to $A_S$ is large.

### 3.1 EFFICIENT ALGORITHM

Our full algorithm is described in Algorithm 1. We will now describe the series of steps in our proof. Let $S$ be the subset of rows that have been selected by the algorithm and $a_{t+1}$ is the row

---

**Algorithm 1:** Minimize Distance to a Subspace

---

**Input:** A matrix $A$ as a stream of rows $a_1, \ldots, a_n \in \mathbb{R}^d$, a rank parameter $k$
**Output:** A subset $S \subseteq [n]$
1 $S \leftarrow \emptyset, \lambda \leftarrow 0$                 `// Algorithm stores` $A_S$
2 **for** $t = 1, \ldots, n$ **do**
3      **if** $\lambda = 0$ **then**
4          **if** $a_t \notin$ *rowspace*$(A_S)$ *or* $a_t^\top (A_S^\top A_S)^+ a_t \geq 1/(1 + 1/k)$ **then**
5              $\lfloor \; S \leftarrow S \cup \{ t \}$
6      **else**
7          **if** $a_t^\top (A_S^\top A_S + \lambda I)^+ a_t \geq 1/(1 + 1/k)$ **then**
8              $\lfloor \; S \leftarrow S \cup \{ t \}$
9      $\lambda \leftarrow \|A_S - [A_S]_k\|_{\mathsf{F}}^2 / k$         `// Changes only when` $S$ `is updated`
10 **return** $S$

---

being processed. The following lemma shows that if there exists a rank $k$ projection $P$ such that $\|a_{t+1}^\top (I - P)\|_2 > \|A_S(I - P)\|_{\mathsf{F}}$, then the algorithm adds $a_{t+1}$ to $S$.

**Lemma 3.1.** *Let $t$ be arbitrary and let $S_t$ be the subset of rows selected by Algorithm 1 after processing the rows $a_1, \ldots, a_t$. If there exists a rank $k$ projection $P$ such that $\|a_{t+1}^\top (I - P)\|_2 > \|A_{S_t}(I - P)\|_{\mathsf{F}}$, then the algorithm adds the row $t + 1$ to the set $S$ that it maintains.*

We use the following fact: if $a_i \in$ rowspace$(B)$, then $\max_{x:Bx \neq 0} |\langle a_i, x\rangle|^2 / \|Bx\|_2^2 = a_i^\top (B^\top B)^+ a_i$ where $A^+$ denotes the Moore-Penrose pseudoinverse of $A$. We then argue that if there exists a rank-$k$ projection matrix $P$ with $\|a_{t+1}^\top (I - P)\|_2 > \|A_S(I - P)\|_{\mathsf{F}}$, then taking $x^* = (I - P)a_{t+1}$, we obtain that $\frac{|\langle a_{t+1}, x\rangle|^2}{\|x^*\|_2^2 \|A_S - [A_S]_k\|_{\mathsf{F}}^2 / k + \|A_S x^*\|_2^2} \geq 1/(1 + 1/k)$ which then implies $a_{t+1}^\top (A_S^\top A_S + I \cdot \|A_S - [A_S]_k\|_{\mathsf{F}}^2 / k)^+ a_{t+1} \geq 1/(1 + 1/k)$ which is exactly the condition in which the algorithm adds the row $t + 1$ to $S$.

From our discussion above, all we need to show now is that the size of the set $S$ is small. We first define online rank-$k$ ridge leverage scores. Given a matrix $A$ and a rank parameter $k$, define the *online rank-$k$ ridge leverage score* of the $i$-th row $a_i$ of $A$, denoted by $\tau_i^{\mathsf{OL},k}(A)$ as follows:

$$
\begin{cases}
1 & \text{rank}(A_{1:i-1}) \leq k \text{ and } a_i \notin \text{rowspace}(A_{1:i-1}) \\
\min(1, a_i^\top (A_{1:i-1}^\top A_{1:i-1})^+ a_i) & \text{rank}(A_{1:i-1}) \leq k \text{ and } a_i \in \text{rowspace}(A_{1:i-1}) \\
\min(1, a_i^\top (A_{1:i-1} A_{1:i-1}^\top + \frac{\|A_{1:i-1} - [A_{1:i-1}]_k\|_{\mathsf{F}}^2}{k} I)^+ a_i) & \text{rank}(A_{1:i}) > k
\end{cases}
$$

Inspecting the algorithm, we see that it is evaluating the rank $k$ online ridge leverage score of $a_{t+1}$ with respect to $A_S$ and if it is greater than $1/(1 + 1/k)$, the algorithm adds $a_{t+1}$ to $S$ and proceeds. Thus, we directly have that the rank-$k$ online ridge leverage score of each row of $A_S$ is at least $1/(1 + 1/k)$. Now, the following lemma bounds the sum of rank-$k$ online ridge leverage scores of any matrix with bounded integer coordinates.

**Lemma 3.2** (Sum of rank-$k$ ridge leverage scores). *Let $A \in \mathbb{R}^{n \times d}$ be an arbitrary matrix with integer entries bounded in absolute value by* $\text{poly}(n)$. *Then the sum of online rank-$k$ ridge leverage scores of the rows of $A$ is at most $O(k^3 \log^2 n)$.*

The proof of this lemma is largely similar to that of Braverman et al. (2020). Hence, the sum of online rank-$k$ ridge leverage scores of any $n \times d$ matrix with integer coordinates bounded in absolute value by $\text{poly}(n)$ is at most $O(k^3 \log^2 n)$. Since we assumed that the entries of the original matrix are bounded in absolute value by $\text{poly}(n)$, we have that the entries of the coreset $A_S$ are also bounded in absolute value by $\text{poly}(n)$. Therefore, the coreset construction algorithm selects at most $O(k^3 \log^2 n)$ rows. Thus, we have the following theorem.

**Theorem 3.3.** *Given rows of any arbitrary $n \times d$ matrix $A$ with integer coordinates bounded in absolute value by* $\text{poly}(n)$, *Algorithm 1 selects a subset $S$ of size $|S| \leq O(k^3 \log^2 n)$ such that for any $k$ dimensional subspace $V$, we have*

$$\|A_S(I - \mathbb{P}_V)\|_{\infty,2} \leq \|A(I - \mathbb{P}_V)\|_{\infty,2} \leq C k^{3/2} \log n \|A(I - \mathbb{P}_V)\|_{\infty,2}$$

*for a large enough constant $C$. Additionally, the space requirement of the algorithm is bounded by $O(d \cdot k^3 \log^3 n)$ bits.*

**Remark 3.4.** In Algorithm 1, the above theorem shows that the set $S$ is updated at most $O(k^3 \log^2 n)$ times. When the set $S$ is updated we recompute the singular value decomposition of $A_S$ to obtain $U_S \Sigma_S V_S^\top$ where $\Sigma_S$ has at most $|S|$ nonzero entries. Now define $\lambda_S$ as $\|A_S - [A_S]_k\|_F^2/k$ and we then have that $(A_S^\top A_S + \lambda_S I)^{+/2} = U(\Sigma_S^2 + \lambda_S)^{-1/2} U^\top + (1/\sqrt{\lambda_S})(I - UU^\top)$. Now, $a_i^\top (A_S^\top A_S + \lambda_S I)^+ a_i = \|(A_S^\top A_S + \lambda_S I)^{+/2} a_i\|_2^2$. When we update the set $S$, we sample a Gaussian matrix $\mathbf{G}$ with $O(\log n)$ rows and then approximate $\|(A_S^\top A_S + \lambda_S I)^{+/2} a_i\|_2^2$ with $\|\mathbf{G}(A_S^\top A_S + \lambda_S I)^{+/2} a_i\|_2^2$. By a union bound, for all rows $a_i$ that appear after $S$ is updated, we have

$$\|\mathbf{G}(A_S^\top A_S + \lambda_S I)^{+/2} a_i\|_2^2 = (1 \pm 1/5)\|(A_S^\top A_S + \lambda_S I)^{+/2} a_i\|_2^2.$$

Now, we modify the algorithm to instead select rows for which $\|\mathbf{G}(A_S^\top A_S + \lambda_S I)^{+/2} a_i\|_2^2 \geq 3/5$. Note that conditioned on above, we still have all the properties of Algorithm 1. Additionally each row can be processed in $O(\mathrm{nnz}(a_i) \cdot \log n)$ if the row is not added to $S$. Hence, the overall running time of the algorithm is $O(\mathrm{nnz}(A) \log n + d \cdot \mathrm{poly}(k, \log n))$.

**Remark 3.5.** Once we compute the coreset, we can use algorithms from earlier works such as Varadarajan et al. (2007) and their approximation ratio translates accordingly to the original matrix $A$ by the above theorem. Since, $A_S$ only has $O(k^3 \log^2 n)$ rows, even the top-$k$ right singular subspace of $A_S$ is a good solution for $\ell_\infty$ approximation. Concretely, let $V^*$ be the optimal $\ell_\infty$ Subspace approximation solution for $A$ and $\tilde{V}$ be the optimal solution for $A_S$. Let $V_k$ be the top $k$ right singular subspace of $A_S$. We have

$$\|A_S(I - V_k V_k^\top)\|_{\infty,2} \leq \|A_S(I - V_k V_k^\top)\|_F \leq \|A_S(I - \tilde{V}\tilde{V}^\top)\|_F \leq \sqrt{|S|}\|A_S(I - \tilde{V}\tilde{V}^\top)\|_{\infty,2}$$

and $\|A_S(I - \tilde{V}\tilde{V}^\top)\|_{\infty,2} \leq \|A_S(I - V^*(V^*)^\top)\|_{\infty,2} \leq \|A(I - V^*(V^*)^\top)\|_{\infty,2}$. Hence, $\|A(I - V_k V_k^\top)\|_{\infty,2} \leq Ck^{3/2} \log n \sqrt{|S|}\|A(I - V^*(V^*)^\top)\|_{\infty,2} = O(k^3 \log^2 n)\|A(I - V^*(V^*)^\top)\|_{\infty,2}$. Thus, we can obtain an $O(k^3 \log^2 n)$-multiplicative approximate solution *without* using any SDP based algorithms from previous works. We can additionally initialize an alternating minimization algorithm on the coreset for $\ell_\infty$ subspace approximation using the SVD subspace of the coreset and use convex optimization solvers to further improve the quality of the solution. We do note that there are no known bounds on the solution quality attained by the alternating minimization algorithm.

By a simple (lossy) reduction of outer $(d - k)$ radius estimation problem to computing optimal $\ell_\infty$ subspace approximation of the matrix $B = A - a_1$ i.e., the matrix obtained by subtracting $a_1$ from each row of $A$, we obtain the following theorem.

**Theorem 3.6** (Outer $(d - k)$ radius estimation). *There is a streaming algorithm, which given an insertion only stream of $d$ dimensional vectors $a_1, \ldots, a_n$ with integer coordinates bounded in absolute value by $\mathrm{poly}(n)$, uses $O(d \cdot k^3 \log^2 n)$ bits of space and outputs an $O(k^{3/2} \log n)$ approximation to the outer $(d - k)$ radius of the points $\{a_1, \ldots, a_n\}$.*

## 4 $\ell_p$ SUBSPACE APPROXIMATION

We now show that our coreset construction algorithm for the $\ell_\infty$ subspace approximation problem, extends to the $\ell_p$ subspace approximation problem. Fix a matrix $A$. For any $k$-dimensional subspace $V$, let $d_V$ denote the nonnegative vector satisfying $(d_V)_i = \mathrm{dist}(a_i, V) = \|a_i^\top(I - \mathbb{P}_V)\|_2$. Hence, the $\ell_p$ subspace approximation problem is to find the rank-$k$ subspace $V$ that minimizes $\|d_V\|_p$. We use exponential random variables to embed $\ell_p$ low rank approximation problem into an $\ell_\infty$ low rank approximation problem. We then use the coreset construction algorithm for $\ell_\infty$ LRA to obtain a coreset for the $\ell_p$ LRA. First, we have the following lemma about exponential random variables that has been used in various previous works to embed $\ell_p$ problems into an $\ell_\infty$ problem.

**Lemma 4.1.** *Let $\mathbf{e}_1, \ldots, \mathbf{e}_n$ be independent exponential random variables. Then with probability $\geq 1 - \delta$, $\max_i \lceil \mathbf{e}_i^{-1/p} \rceil |x_i| \geq \|x\|_p/(\log 1/\delta)^{1/p}$. We also have that with probability $\geq 1 - \delta$, $\max_i \lceil \mathbf{e}_i^{-1/p} \rceil |x_i| \leq (\delta^{-1/p} + 1)\|x\|_p$.*

Given $n$, define $\mathbf{D}$ to be a random matrix with diagonal entries given by independent copies of the random variable $\lceil \mathbf{e}^{-1/p} \rceil$. For any fixed rank $k$ projection matrix $P$, the above lemma implies that

$\|\mathbf{D}A(I-P)\|_{\infty,2} \geq \|A(I-P)\|_{p,2}/(\log 1/\delta)^{1/p}$. But we can not union bound over the net of all $k$ dimensional subspace of $\mathbb{R}^d$ since the net can have as many as $\exp(dk)$ subspaces which leads to a distortion of $d^{1/p}$ which is prohibitive. Here we crucially use the fact that Algorithm 1 only selects a coreset with $K = O(k^3 \log^2 n)$ rows. So only those $k$ dimensional subspaces spanned by at most $K$ rows of $A$ are interest to us. Now, we can union bound over a net of $\exp(\text{poly}(k) \log n)$ subspaces and show the following lemma:

**Lemma 4.2.** *Let* $\mathbf{D}$ *be an* $n \times n$ *diagonal matrix with each diagonal entry being an independent copy of the random variable* $\lceil e^{-1/p} \rceil$. *Fix an* $n \times d$ *matrix A. With probability* $\geq 98/100$, *for all* $k$ *dimensional subspaces that are in the span of at most* $m = O(k^3 \log^2 n)$ *rows of A, we have,*

$$\|\mathbf{D} \cdot d_V\|_\infty \geq \|d_V\|_p/2(\log 100 + m \log n + k^2 m \log n)^{1/p}.$$

If $V^*$ is the optimal solution for the $\ell_p$ subspace approximation problem, we can also condition on the event that $\|\mathbf{D} \cdot d_{V^*}\|_\infty \leq C\|d_{V^*}\|_p$ for a large enough constant $C_1$.

We can now argue that if $S$ is the subset of rows selected by Algorithm 1 when run on the matrix $\mathbf{D}A$,

$$\hat{V} \doteq \underset{k\text{-dim } V}{\arg\min} \|(\mathbf{D}A)_S(I - \mathbb{P}_V)\|_{\infty,2}$$

is also a good solution for the $\ell_p$ Subspace Approximation problem as follows: We first note that $\hat{V}$ is a $k$-dimensional subspace in the rowspace of $A_S$. Hence, $\|d_{\hat{V}}\|_p \leq O(k^{5/p} \log^{3/p} n)\|d_{\hat{V}}\|_\infty$. Using Theorem 3.3, we have $\|d_{\hat{V}}\|_\infty = \|\mathbf{D}A(I - \mathbb{P}_{\hat{V}})\|_{\infty,2} \leq Ck^{3/2} \log n\|(\mathbf{D}A)_S(I - \mathbb{P}_{\hat{V}})\|_{\infty,2}$. Since $\hat{V}$ is optimal for the matrix $(\mathbf{D}A)_S$, we have $\|(\mathbf{D}A)_S(I - \mathbb{P}_{\hat{V}})\|_{\infty,2} \leq \|(\mathbf{D}A)_S(I - \mathbb{P}_{V^*})\|_{\infty,2}$. Now, $\|(\mathbf{D}A)_S(I - \mathbb{P}_{V^*})\|_{\infty,2} \leq \|\mathbf{D}A(I - \mathbb{P}_{V^*})\|_{\infty,2} \leq C_1\|A(I - \mathbb{P}_{V^*})\|_{p,2}$. Thus overall, we have $\|d_{\hat{V}}\|_p \leq O(k^{5/p+3/2} \log^{1+3/p} n)\|d_{V^*}\|_p$. giving the following theorem.

**Theorem 4.3.** *Let* $\mathbf{D}$ *be an* $n \times n$ *random matrix with each diagonal entry being an independent copy of* $\lceil e^{-1/p} \rceil$ *where* e *is a standard exponential random variable. If* $S$ *is the subset selected by Algorithm 1 when run on the rows of the matrix* $\mathbf{D} \cdot A$ *and if* $\hat{V}$ *is the optimal solution to the problem* $\min_{k\text{-dim } V} \|(\mathbf{D}A)_S(I - \mathbb{P}_V)\|_{\infty,2}$, *then with probability* $\geq 9/10$,

$$\|A(I - \mathbb{P}_{\hat{V}})\|_{p,2} \leq O(k^{3/2+5/p} \log^{1+3/p} n) \min_{k\text{-dim } V} \|A(I - \mathbb{P}_V)\|_{p,2}.$$

## 5 APPLICATIONS TO OTHER GEOMETRIC STREAMING PROBLEMS

Given a matrix $A$, suppose that the rows of $A$ are close to a $k$-dimensional subspace in the following sense: $\Delta \doteq \min_{k\text{-dim } V} \|A(I - \mathbb{P}_V)\|_{\infty,2}$ is small. We now show that if $S$ is the subset of rows selected by Algorithm 1, then for any vector $x$, $\|Ax\|_\infty$ can be approximated using $\|A_Sx\|_\infty$. Fix any unit vector $x$. Let $i$ be the index such that $\|Ax\|_\infty = |\langle a_i, x \rangle|$. If $i \in S$, we clearly have $\|Ax\|_\infty = \|A_Sx\|_\infty$. If $i \notin S$, by proof of Lemma 3.1, we obtain that

$$\max_x \frac{|\langle a_i, x \rangle|^2}{\|A_{S<i}x\|_2^2 + \|A_{S<i} - [A_{S<i}]_k\|_{\mathsf{F}}^2/k} \leq \frac{1}{1+1/k}$$

which implies $\|Ax\|_\infty^2 = |\langle a_i, x \rangle|^2 \leq \|A_{S<i}x\|_2^2 + \|A_{S<i} - [A_{S<i}]_k\|_{\mathsf{F}}^2/k \leq \|A_Sx\|_2^2 + \|A_S - [A_S]_k\|_{\mathsf{F}}^2/k$. Let $V^*$ be the optimal solution for rank-$k$ $\ell_\infty$ subspace approximation of $A$. We then have, $\|Ax\|_\infty^2 \leq \|A_Sx\|_2^2 + \|A_S(I - \mathbb{P}_{V^*})\|_{\mathsf{F}}^2/k \leq \|A_Sx\|_2^2 + |S|\Delta^2/k$. Using $|S| = O(k^3 \log^2 n)$, we get the following lemma.

**Lemma 5.1.** *If* $S$ *is the subset of rows selected by Algorithm 1, for any* $k$-*dimensional subspace* $U$ *and any unit vector* $x$,

$$\|A_Sx\|_2/Ck^{3/2} \log n \leq \|A_Sx\|_\infty \leq \|Ax\|_\infty \leq \|A_Sx\|_2 + (Ck \log n)\Delta. \tag{1}$$

*Additionally, as* $\|A_Sx\|_2 \leq \sqrt{|S|}\|A_Sx\|_\infty$, *we also have*

$$\|A_Sx\|_\infty \leq \|Ax\|_\infty \leq (Ck^{3/2} \log n)\|A_Sx\|_\infty + (Ck \log n)\Delta. \tag{2}$$

**Width Estimation.** Given a point set $a_1, \ldots, a_n \in \mathbb{R}^d$, then the width of the point set in the direction $x \in \mathbb{R}^d$, for a unit vector $x$ is defined as $w(x) \doteq \max_i \langle a_i, x \rangle - \min_i \langle a_i, x \rangle$. Using a coreset for estimating $\|Ax\|_\infty$, Woodruff and Yasuda (2022) give an $O(\sqrt{d \log n})$ approximation to the width estimation problem. Using Lemma 5.1, we show that we get better approximations when $\Delta$ is small.

We can equivalently write $w(x) = \max_i \langle a_i - a_1, x \rangle - \min_i \langle a_i - a_1, x \rangle$. Now, $\max_i \langle a_i - a_1, x \rangle \geq \langle 0, x \rangle = 0$ and $\min_i \langle a_i - a_1, x \rangle \leq \langle 0, x \rangle \leq 0$ which implies that $\|(A - a_1)x\|_\infty \leq w(x) \leq 2\|(A - a_1)x\|_\infty$. If $S$ is the subset selected by the algorithm when run on the rows $0 = a_1 - a_1, a_2 - a_1, \ldots, a_n - a_1$, then from Lemma 5.1, we have $\|(A - a_1)_S x\|_\infty \leq \|(A - a_1)x\|_\infty \leq w(x)$ and $w(x) \leq 2\|(A - a_1)x\|_\infty \leq 2Ck^{3/2} \log n \|(A - a_1)_S x\|_\infty + 2Ck \log n \Delta$. Thus, $w'(x) \doteq \|(A - a_1)_S x\|_\infty$ satisfies

$$w(x)/2Ck^{3/2} \log n - \Delta/\sqrt{k} \leq w'(x) \leq w(x)$$

for a large enough constant $C$. When $\Delta$ is very small, for the interesting directions where width is large enough, we obtain a better multiplicative error of $O(k^{3/2} \log n)$ as compared to $O(\sqrt{d \log n})$ achieved by the algorithm of Woodruff and Yasuda (2022).

**$r$-Robust Directional Width.** The presence of some outliers in the direction $x$ distorts the width in the direction $x$ by a lot. So a more robust version, parameterized by a positive integer $r$, called $r$-robust directional width is studied. It is defined to be the $r$-th largest value among the set $\{|\langle a_i, x \rangle| \mid i \in [n]\}$. The $r$-robust directional width in the direction $x$ of a matrix $A$ with rows given $a_1, \ldots, a_n$ is denoted by $\mathcal{E}_r(x, A)$.

Using the *peeling* technique of Agarwal et al. (2008), Woodruff and Yasuda (2022) give an algorithm for approximating the $r$-Robust Directional Width using their coreset. The same technique directly implies that we can construct a coreset with $O(r \cdot k^3 \log^2 n)$ rows such that for every $x$,

$$\mathcal{E}_r(x, A)/Ck^{3/2} \log n - \Delta/\sqrt{k} \leq \mathcal{E}_r(x, A_S) \leq \mathcal{E}_r(x, A)$$

If $\mathcal{E}_r(x, A)$ is large enough compared to $\Delta$, then $\mathcal{E}_r(x, A_S)$ is a better approximation for $\mathcal{E}_r(x, A)$ than the $O(\sqrt{d \log n})$ approximation guaranteed by Woodruff and Yasuda (2022).

**Löwner-John Ellipsoid.** Given a symmetric convex body, the Löwner-John Ellipsoid is defined to be the ellipsoid of minimum volume that encloses the convex body. We consider the case when the convex body is defined as $K = \{x \mid \|Ax\|_\infty \leq 1\}$ where the streaming algorithm sees the rows of matrix $A$ one after the other. Woodruff and Yasuda (2022) show that their coreset can be used to compute an ellipsoid $E'$ such that $E' \subseteq K \subseteq O(\sqrt{d \log n})E'$ thereby obtaining a $O(d\sqrt{\log n})$-approximate Löwner-John Ellipsoid.

When $k \ll d$, Algorithm 1 selects $\ll d$ number of rows and does not have the full $d$-dimensional *view* of the point set and hence can not compute an ellipsoid that satisfies the above definition if the points spans $\mathbb{R}^d$. So we consider the set $K \cap B(0, 1)$ and give an algorithm that computes an unbounded ellipsoid $E'$ such that $E' \cap B(0, 1) \subseteq K \cap B(0, 1) \subseteq (\alpha E') \cap B(0, 1)$.

By Lemma 5.1, we have that if $\|Ax\|_\infty \leq 1$ and $\|x\|_2 = 1$, then $\|A_S x\|_2 \leq Ck^{3/2} \log n$ and if $\|A_S x\|_2 \leq 1 - (Ck \log n)\Delta$ and $\|x\|_2 \leq 1$, then $\|Ax\|_\infty \leq 1$. Now assuming $\Delta < 1/(Ck \log n)$, define $E' = \{x \mid \|A_S x\|_2 \leq 1 - (Ck \log n)\Delta\}$.

Note that when $\text{rank}(A_S) < d$, the set $E'$ is unbounded. From the above, we have that if $x \in E' \cap B(0, 1)$, then $x \in K \cap B(0, 1)$. Additionally if $x \in K \cap B(0, 1)$, then $\|A_S x\|_2 \leq Ck^{3/2} \log n$ and therefore $x \in \frac{Ck^{3/2} \log n}{1 - (Ck \log n)\Delta} E' \cap B(0, 1)$. Hence,
$$E' \cap B(0, 1) \subseteq K \cap B(0, 1) \subseteq \frac{Ck^{3/2} \log n}{1 - (Ck \log n)\Delta} E' \cap B(0, 1).$$

## 6 EXPERIMENTS

We implement our coreset construction algorithm (Algorithm 1) and show that the coreset constructed has a low distortion both for the $\ell_\infty$ low rank approximation and width estimation.

### 6.1 $\ell_\infty$ LOW RANK APPROXIMATION

Theorem 3.3 shows that the coreset computed by Algorithm 1 lets us approximate $\|A(I - \mathbb{P}_V)\|_{\infty,2}$ for any projection matrix $V$. We run our algorithm on a synthetic data set and a real world dataset. We

construct our synthetic dataset as follows: we pick $n = 40,000$, $d = 10,000$ and $k = 20$. We sample an $n \times k$ random matrix $L$ and a $k \times d$ random matrix $R$ each with i.i.d uniform random variables drawn from $\{-100, -99, \ldots, 100\}$. We create an $n \times d$ matrix $A \doteq L \cdot R + G$ where $G$ is a noise matrix with each entry being an i.i.d uniform random variable drawn from $\{-5000, \ldots, 5000\}$. With parameter $k = 20$, when Algorithm 1 is run on the matrix $A$, the coreset $A_S$ computed by the algorithm has only 28 rows. To measure the *quality* of the coreset, we consider the following candidate subspaces: we define $V_i$ to be the at most $i$ dimensional subspace formed by the first $i$ rows of $R$. These are indeed the subspaces for which the rows of $A$ have a *low* distance to. We obtain that

$$1 \leq \max_{i \in [20]} \frac{\|A(I - \mathbb{P}_{V_i})\|_{\infty,2}}{\|A_S(I - \mathbb{P}_{V_i})\|_{\infty,2}} \leq 1.3433$$

which shows that the $\ell_\infty$ cost of the interesting subspaces estimated using the coreset is not too small compared to the actual $\ell_\infty$ cost of the subspace. Another important requirement is that we do not underestimate the cost of uninteresting subspaces by a lot. To see this, we generate random subspaces of $k = 20$ dimensions and observe that $\|A(I - \mathbb{P}_V)\|_{\infty,2}/\|A_S(I - \mathbb{P}_V)\|_{\infty,2} \leq 1.05$ with high probability when $V$ is drawn at random. This can be explained by the fact that random subspaces are so bad in that $\|A(I - \mathbb{P}_V)\|_{\infty,2} \approx \|A\|_{\infty,2}$ since a random subspace does not capture a large part of the row of $A$ with the largest norm. So essentially when $V$ is a random matrix, $\|A(I - \mathbb{P}_V)\|_{\infty,2}/\|A_S(I - \mathbb{P}_V)\|_{\infty,2} = \|A\|_{\infty,2}/\|A_S\|_\infty$ and since all the rows of $A$ have similar norms, we get that $\|A(I - \mathbb{P}_V)\|_{\infty,2}/\|A_S(I - \mathbb{P}_V)\|_{\infty,2} \approx 1$.

For the real world dataset, we consider a grayscale image Leung (2017) of dimensions $1836 \times 3264$ and treat the image as a matrix $A$ of the same dimensions. We observe that a rank 150 approximation of the image computed using the SVD is very close to the original image (with some artifacts) and therefore set $k = 150$ to be the parameter for which we want to solve the $\ell_\infty$ low rank approximation problem. We run the coreset construction algorithm on $A$ and obtain a coreset $A_S$ with 312 rows. Note that the number of rows in the coreset is $\approx 17\%$ of the original matrix. Again, to measure the quality of the coreset, we consider subspace $V_i$ defined to the top $i$ dimensional right singular subspace of $A$ and measure $\|A(I - \mathbb{P}_{V_i})\|_{\infty,2}/\|A_S(I - \mathbb{P}_{V_i})\|$. We obtain $\max_{i \in [k]} \|A(I - \mathbb{P}_{V_i})\|_{\infty,2}/\|A_S(I - \mathbb{P}_{V_i})\|_{\infty,2} \leq 1.09$ and hence the coreset gives very accurate cost estimates for these interesting subspaces. We repeat the same experiment on a different grayscale image European Space Agency and NASA (2006) of dimensions $4690 \times 6000$ and use $k = 200$. We obtain a coreset $A_S$ with 382 rows and for $V_i$ defined in the same way as before, $\max_i \|A(I - \mathbb{P}_{V_i})\|_{\infty,2}/\|A_S(I - \mathbb{P}_{V_i})\|_{\infty,2} \leq 1.12$.

## 6.2 WIDTH ESTIMATION

Towards width estimation, Lemma 5.1 shows that if $A_S$ is the coreset computed by Algorithm 1, then for any unit vector, $\|Ax\|_\infty$ can be approximated up to a multiplicative/additive error. We again consider synthetic/real-world datasets and use linear programs to obtain an upper bound on $\|Ax\|_\infty/\|A_S x\|_\infty$ for $x \in \text{rowspace}(A_S)$. We note that when the rows of $A$ are close to a $k$-dimensional subspace, then $A_S$ computed using Algorithm 1 spans a subspace close to this $k$-dimensional subspace by Theorem 3.3. Hence, all the *important* directions are already in rowspace($A_S$) and bounding $\|Ax\|_\infty/\|A_S x\|_\infty$ for $x \in \text{rowspace}(A_S)$ verifies that the distortion in the important directions is not large.

We construct a synthetic dataset $A = L \cdot R + G$ in a similar way to the previous section with $n = 40,000$, $d = 10,000$ and $k = 20$. To avoid numerical issues when solving linear programs, we now choose the coefficients of the matrices $L$ and $R$ to be i.i.d uniform random variables drawn from $\{-10, \ldots, 10\}$ and the coefficients of $G$ to be i.i.d uniform random variables drawn from $\{-50, \ldots, 50\}$. The coreset $A_S$ constructed by Algorithm 1 for the matrix $A$ has 29 rows and by solving $n$ linear programs, we find that $\max_{x \in \text{rowspace}(A_S)} \|Ax\|_\infty/\|A_S x\|_\infty \leq 4.8$.

We also perform the same experiment on the images from previous section and find that $\|Ax\|_\infty/\|A_S x\|_\infty \leq 1.005$ for all $x \in \text{rowspace}(A_S)$ for the first image and $\|Ax\|_\infty/\|A_S x\|_\infty \leq 1.03$ for all $x \in \text{rowspace}(A_S)$ for the second image. For real-world datasets, the coreset computed is very accurate in approximating $\|Ax\|_\infty$ for all the interesting directions $x$. This can be explained by the fact that the value of $k$ we picked is large and the noise at that value of $k$ is small enough that many directions are *covered* by the coreset and hence the coreset has a very small error in estimating $\|Ax\|_\infty$.

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

# A  MISSING PROOFS FROM SECTION 3

## A.1  PROOF OF LEMMA 3.1

*Proof.* Let $S_t$ be the set $S$ after the algorithm processes rows $a_1, \ldots, a_t$. Suppose $\|A_{S_t}(I-P)\|_{\mathsf{F}} \neq 0$. Define $x^* = (I - P)a_{t+1}/\|(I - P)a_{t+1}\|_2$. Now,

$$|\langle a_{t+1}, x^* \rangle|^2 = \frac{(a_{t+1}^\top (I - P)a_{t+1})^2}{\|(I - P)a_{t+1}\|_2^2} = \|(I - P)a_{t+1}\|_2^2.$$

We also have

$$\|A_{S_t} x^*\|_2^2 = \frac{\|A_{S_t}(I - P)a_{t+1}\|_2^2}{\|(I - P)a_{t+1}\|_2^2} \leq \frac{\|A_{S_t}(I - P)\|_{\mathsf{F}}^2 \|(I - P)a_{t+1}\|_2^2}{\|(I - P)a_{t+1}\|_2^2} = \|A_{S_t}(I - P)\|_{\mathsf{F}}^2.$$

Additionally, when processing the row $a_{t+1}$, we have $\lambda = \|A_{S_t} - [A_{S_t}]_k\|_{\mathsf{F}}^2/k \leq \|A_{S_t}(I - P)\|_{\mathsf{F}}^2/k$ since $P$ is a rank $k$ projection. Overall if either $\lambda \neq 0$ or $A_{S_t} x^* \neq 0$, we have $\lambda \|x^*\|_2^2 + \|A_{S_t} x^*\|_2^2 \neq 0$ and

$$\frac{|\langle a_{t+1}, x^* \rangle|^2}{\lambda \|x^*\|_2^2 + \|A_{S_t} x^*\|_2^2} \geq \frac{\|(I - P)a_{t+1}\|_2^2}{\|A_{S_t}(I - P)\|_{\mathsf{F}}^2/k + \|A_{S_t}(I - P)\|_{\mathsf{F}}^2} \geq \frac{1}{1 + 1/k}$$

where we used the assumption that $\|(I - P)a_{t+1}\|_2 > \|A_{S_t}(I - P)\|_{\mathsf{F}}$. Now, we note that for any matrix $B$ if $a_{t+1} \in \mathrm{rowspace}(B)$,

$$\max_{x:Bx \neq 0} \frac{|\langle a_{t+1}, x \rangle|^2}{\|Bx\|_2^2} = a_{t+1}^\top (B^\top B)^+ a_{t+1}.$$

If $\lambda \neq 0$, then $a_{t+1} \in \mathrm{rowspace}(\lambda I)$ using which we obtain that $a_{t+1}^\top (A_{S_t}^\top A_{S_t} + \lambda I)^+ a_{t+1} \geq 1/(1 + 1/k)$ which implies that the algorithm adds the row $a_{t+1}$ to the set $S$. Suppose that $\lambda = 0$. We then have that $\mathrm{rank}(A_{S_t}) \leq k$. In this case, if $a_{t+1} \notin \mathrm{rowspace}(A_{S_t})$, then the algorithm adds $a_{t+1}$ to $S$ and we are done. If $a_{t+1} \in \mathrm{rowspace}(A_{S_t})$, then we claim that $A_{S_t} x^* \neq 0$. Suppose $A_{S_t} x^* = 0$. Since $a_{t+1} \in \mathrm{rowspace}(A_{S_t})$, by taking appropriate linear combination of the rows of $A_{S_t} x^*$, we obtain that $\langle a_{t+1}, x^* \rangle = 0$ which then implies $\|(I - P)a_{t+1}\|_2 = 0$ which contradicts our assumption. We now have $|\langle a_{t+1}, x^* \rangle|^2/\|A_{S_t} x^*\|_2^2 \geq 1/(1+1/k)$ and since $a_{t+1} \in \mathrm{rowspace}(A_{S_t})$, we obtain that $a_{t+1}^\top (A_{S_t}^\top A_{S_t})^+ a_{t+1} \geq 1/(1 + 1/k)$ and therefore the algorithm adds $a_{t+1}$ to $S$.

Suppose $A_{S_t}(I - P) = 0$. Then $\lambda = \|A_{S_t} - [A_{S_t}]_k\|_{\mathsf{F}}^2/k = 0$ and that $\mathrm{rowspace}(A_{S_t}) \subseteq \mathrm{rowspace}(P)$. Now, $\|a_{t+1}^\top (I - P)\|_2 > 0$ implies that $a_{t+1} \notin \mathrm{rowspace}(A_{S_t})$ and therefore the algorithm adds $a_{t+1}$ to the set $S$. □

## A.2  PROOF OF LEMMA 3.2

First we prove the following lemma.

**Lemma A.1.** *If $A$ is an $n \times d$ matrix with integer entries bounded by $\mathrm{poly}(n)$ and $\mathrm{rank}(A) \geq t$, then $\sigma_t(A) \geq (\mathrm{poly}(n))^{-(t-1)/2}$.*

*Proof.* Let $\mathrm{rank}(A) = r$ and note that $\sigma_i(A)^2 = \lambda_i(A^\top A)$ where $\lambda_i$ denotes the $i$-th largest eigenvalue of the matrix $A^\top A$. We have that the nonzero roots of the degree $d$ polynomial $\det(A^\top A - \lambda I) = 0$ are exactly equal to $\lambda_1(A^\top A), \ldots, \lambda_r(A^\top A)$ and that

$$\det(A^\top A - \lambda I) = (\lambda_1(A^\top A) - \lambda_1) \cdots (\lambda_r(A^\top A) - \lambda)\lambda^{d-r}.$$

Hence the coefficient of $\lambda^{d-r}$ in the polynomial $\det(A^\top A - \lambda I)$ is exactly equal to $(-1)^r \prod_{i=1}^r \lambda_i(A^\top A) \neq 0$. We now observe that the entries of the matrix $A^\top A$ are all integers and therefore all the coefficients in the polynomial $\det(A^\top A - \lambda I)$ are also integers. Hence,

$$\prod_{i=1}^r \lambda_i(A^\top A) \neq 0 \implies \prod_{i=1}^r \lambda_i(A^\top A) \geq 1.$$

Now we either have $\lambda_t(A^\top A) \geq 1$ in which case we are done or $\lambda_t(A^\top A) < 1$. From here on assume for now that $\lambda_t(A^\top A) < 1$ which implies $\prod_{i=1}^t \lambda_i(A^\top A) \geq \prod_{i=1}^r \lambda_i(A^\top A) \geq 1$. We now have $\lambda_1(A^\top A) \leq \|A^\top A\|_\mathsf{F} \leq \mathrm{poly}(n)$ using which we obtain

$$\lambda_t(A^\top A) \geq \frac{1}{\lambda_1(A^\top A) \cdots \lambda_{t-1}(A^\top A)} \geq \frac{1}{(\mathrm{poly}(n))^{t-1}}.$$

Hence, $\sigma_t(A) = \sqrt{\lambda_t(A^\top A)} \geq (\mathrm{poly}(n))^{-(t-1)/2}$. □

Now we prove Lemma 3.2

*Proof.* Let $i^* \in [n]$ be the largest index such that $\mathrm{rank}(A_{1:i^*}) = k$. If no such index exists, then set $i^* = n$. For all $i \leq i^* + 1$, by definition, we have that $\tau_i^{\mathsf{OL},k}(A)$ is exactly equal to the online leverage score of the row $i$. Since $\mathrm{rank}(A_{1:i^*+1}) = k + 1$, by Theorem 1.5 of Woodruff and Yasuda (2022), we obtain that

$$\sum_{i=1}^{i^*+1} \tau_i^{\mathsf{OL},k}(A) \leq O(k \log n).$$

We now bound $\sum_{i=i^*+2}^n \tau_i^{\mathsf{OL},k}(A)$. Note that for all $i \geq i^* + 2$,

$$\|A_{1:i-1} - [A_{1:i-1}]_k\|_\mathsf{F}^2 > 0$$

since $\mathrm{rank}(A_{1:i-1}) > k$. Now define $\lambda_i \doteq \|A_{1:i} - [A_{1:i}]_k\|_\mathsf{F}^2/k$ and partition the interval $[\lambda_{i^*+1}, \lambda_n]$ into intervals of the form $[2^j \lambda_{i^*+1}, 2^{j+1}\lambda_{i^*+1})$ for $j = 0, 1, \dots$. Note that there are at most $O(\log(\lambda_n/\lambda_{i^*+1}))$ such intervals. Define $\lambda^{(j)} = 2^j \lambda_{i^*+1}$ and note that if $\lambda^{(j)} \leq \lambda_{i-1} < \lambda^{(j+1)}$ then

$$a_i^\top (A_{1:i-1}^\top A_{1:i-1} + \lambda_{i-1}I)^+ a_i \leq 2a_i^\top (A_{1:i-1}^\top A_{1:i-1} + \lambda^{(j+1)}I)^+ a_i.$$

Hence,

$$\sum_{i:\lambda^{(j)} \leq \lambda_{i-1} < \lambda^{(j+1)}} \tau_i^{\mathsf{OL},k}(A) \leq 2 \sum_{i:\lambda^{(j)} \leq \lambda_{i-1} < \lambda^{(j+1)}} \max(1, a_i^\top (A_{1:i-1}^\top A_{1:i-1} + \lambda^{(j+1)}I)^+ a_i)$$

Let $i_j$ be the largest index such that $\lambda_{i_j} < \lambda^{(j+1)}$. Hence,

$$\sum_{i:\lambda^{(j)} \leq \lambda_{i-1} < \lambda^{(j+1)}} \max(1, a_i^\top (A_{1:i-1}^\top A_{1:i-1} + \lambda^{(j+1)}I)^+ a_i) \leq 1 + \sum_{i=1}^{i_j} \max(1, a_i^\top (A_{1:i-1}^\top A_{1:i-1} + \lambda_{i_j}I)^+ a_i).$$

From the proof of Lemma 2.11 of Braverman et al. (2020), we obtain that

$$\sum_{i=1}^{i_j} \max(1, a_i^\top (A_{1:i-1}^\top A_{1:i-1} + \lambda_{i_j}I)^+ a_i) = O\left(k \log \frac{k\|A_{1:i_j}\|_\mathsf{F}^2}{\|A_{1:i_j} - [A_{1:i_j}]_k\|_\mathsf{F}^2}\right).$$

From the assumption that the entries of matrix $A$ are bounded by $\mathrm{poly}(n)$, we get $\|A_{1:i_j}\|_\mathsf{F}^2 \leq \mathrm{poly}(n)$. Now,

$$\|A_{1:i_j} - [A_{1:i_j}]\|_\mathsf{F}^2 \geq \sigma_{k+1}(A_{1:i_j})^2.$$

Using Lemma A.1, we obtain that $\|A_{1:i_j} - [A_{1:i_j}]_k\|_\mathsf{F}^2 \geq 1/(\mathrm{poly}(n))^{k/2}$ and therefore have

$$k \log \frac{k\|A_{1:i_j}\|_\mathsf{F}^2}{\|A_{1:i_j} - [A_{1:i_j}]_k\|_\mathsf{F}^2} \leq O(k^2 \log n).$$

We similarly have $O(\log \lambda_n/\lambda_{i^*+1}) \leq O(k \log n)$ and therefore there are at most $O(k \log n)$ intervals into which $[\lambda_{i^*+1}, \lambda_n]$ is partitioned which overall implies

$$\sum_{i=i^*+2}^n \tau_i^{\mathsf{OL},k}(A) \leq O(k^3 \log^2 n).$$ □

### A.3 PROOF OF THEOREM 3.6

*Proof.* If $V$ is a $k$-dimensional subspace and $c$ is arbitrary, then the set $V + c$ is defined as a $k$-dimensional flat. Recall that the outer $d - k$ radius of a point set $\{a_1, \ldots, a_n\} \subseteq \mathbb{R}^d$ is defined as

$$\min_{k\text{-dim flat } F} \max_i d(a_i, F).$$

Using the fact that flats are translations of $k$ dimensional subspaces, we equivalently have that the outer $d - k$ radius is equal to

$$\min_{k\text{-dim subspace } V} \min_{c \in \mathbb{R}^d} \max_i d(a_i - c, V) = \min_{k\text{-dim subspace } V} \min_c \|(A - c)(I - \mathbb{P}_V)\|_{\infty, 2}.$$

Here we abuse the notation and use $A - c$ to denote the matrix with rows given by $a_i - c$ for $i \in [n]$. Now define a matrix $B \doteq A - a_1$ with $n$ rows given by $0 = a_1 - a_1, a_2 - a_1, a_3 - a_2, \ldots, a_n - a_1$. For any $k$-dimensional subspace $V$ and any $c \in \mathbb{R}^d$, we have

$$
\begin{aligned}
\|B(I - \mathbb{P}_V)\|_{\infty, 2} = \|(A - a_1)(I - \mathbb{P}_V)\|_{\infty, 2} &= \|(A - c + c - a_1)(I - \mathbb{P}_V)\|_{\infty, 2} \\
&\leq \|(A - c)(I - \mathbb{P}_V)\|_{\infty, 2} + \|(I - \mathbb{P}_V)(a_1 - c)\|_2 \\
&\leq 2\|(A - c)(I - \mathbb{P}_V)\|_{\infty, 2}.
\end{aligned}
$$

Hence, $\|B(I - \mathbb{P}_V)\|_{\infty, 2} \leq 2\min_c \|(A - c)(I - \mathbb{P}_V)\|_{\infty, 2}$. We also have $\|B(I - \mathbb{P}_V)\|_{\infty, 2} = \|(A - a_1)(I - \mathbb{P}_V)\|_{\infty, 2} \geq \min_c \|(A - c)(I - \mathbb{P}_V)\|_{\infty, 2}$. Thus, $\min_V \|B(I - \mathbb{P}_V)\|_{\infty, 2}$ is a 2-approximation for $\min_{k\text{-dim flat } F} \max_i d(a_i, F)$ and if $S$ is the set of rows selected by Algorithm 1 when run on the rows of the matrix $B = A - a_1$, then

$$\min_V \|B_S(I - \mathbb{P}_V)\|_{\infty, 2}$$

is an $O(k^{3/2} \log n)$ approximation for outer $(d - k)$-radius estimation of the point set $\{a_1, \ldots, a_n\}$. $\square$

## B MISSING PROOFS FROM SECTION 4

### B.1 PROOF OF LEMMA 4.1

*Proof.* By min-stability of exponential random variables, we have that the distribution of $\max_i \mathbf{e}^{-1}|x_i|^p$ is the same as the distribution of $\mathbf{e}^{-1}\|x\|_p^p$ where $\mathbf{e}$ is also a standard exponential random variable. With probability $\geq 1 - \delta$, we have $\mathbf{e} \leq \log 1/\delta$. And hence we have that with probability $\geq 1 - \delta$,

$$\max_i \mathbf{e}_i^{-1/p}|x_i| = (\max_i \mathbf{e}_i^{-1}|x_i|^p)^{1/p} \geq \frac{\|x\|_p}{(\log 1/\delta)^{1/p}}.$$

As $\lceil \mathbf{e}_i^{-1/p} \rceil \geq \mathbf{e}_i^{-1/p}$, we have that with probability $\geq 1 - \delta$, $\max_i \lceil \mathbf{e}_i^{-1/p} \rceil |x_i| \geq \|x\|_p/(\log 1/\delta)^{1/p}$. With probability $\geq 1 - \delta$, we also have that $\mathbf{e} \geq \delta$ which implies that with probability $\geq 1 - \delta$, $\max_i \mathbf{e}_i^{-1/p}|x_i| = (\max_i \mathbf{e}_i^{-1}|x_i|^p)^{1/p} \leq \|x\|_p \delta^{-1/p}$. Conditioned on this event, we have $\max_i \lceil \mathbf{e}_i^{-1/p} \rceil |x_i| \leq \max_i (\mathbf{e}_i^{-1/p} + 1)|x_i| \leq \max_i \mathbf{e}_i^{-1/p}|x_i| + \|x\|_\infty \leq (1 + \delta^{-1/p})\|x\|_p$. $\square$

### B.2 PROOF OF LEMMA 4.2

*Proof.* Let $S$ be an arbitrary set of $m \doteq O(k^3 \log^2 n)$ rows of $A$ and let $V_S \doteq \text{rowspace}(A_S)$. Let $N_S$ be a $\gamma$ net for the set $V_S \cap \mathbb{S}^{d-1}$ i.e., the set of vectors in the subspace $V_S$ with euclidean norm 1. As the subspace $V_S$ has dimension at most $m$, we have that there is a set $N_S$ with size at most $\exp(O(m \log 1/\gamma))$. Let $V$ be an arbitrary $k$ dimensional subspace of $V_S$ and let $\{v_1, \ldots, v_k\}$ be an orthonormal basis for $V$.

Let $\tilde{V}$ be the subspace spanned by $\{\tilde{v}_1, \ldots, \tilde{v}_k\}$, where $\tilde{v}_i \in N_S$ and $\|v_i - \tilde{v}_i\|_2$ for all $i \in [n]$. Let $a$ be an arbitrary vector. By abusing the notation let $V$ (resp. $\tilde{V}$) also denote the matrix with $v_1, \ldots, v_k$ (resp. $\tilde{v}_1, \ldots, \tilde{v}_k$) as columns. We have

$$d(a, V) = \|a - VV^\top a\|_2 \quad \text{and} \quad d(a, \tilde{V}) = \|a - \tilde{V}\tilde{V}^+ a\|_2$$

and therefore $|d(a, V) - d(a, \tilde{V})| \leq \|\tilde{V}\tilde{V}^+ - VV^\top\|_2\|a\|_2$. If $\gamma \leq 1/4\sqrt{k}$, we can show that $\|VV^\top - \tilde{V}\tilde{V}^+\|_2 \leq 4\sqrt{k}\gamma$ and therefore have that for any $a$, $|d(a, V) - d(a, \tilde{V})| \leq \sqrt{k}\gamma\|a\|_2$. Hence,

$$\|d_V - d_{\tilde{V}}\|_\infty \leq \max_i |d(a_i, V) - d(a_i, \tilde{V})| \leq 4\sqrt{k}\gamma \max_i \|a_i\|_2 = 4\sqrt{k}\gamma\|A\|_{\infty,2}.$$

Overall, this implies that for any arbitrary $k$ dimensional subspace $V$ in the span of rows of $A_S$, there is a $k$ dimensional subspace $\tilde{V}$ spanned by some $k$ vectors in the net $N_S$ satisfying

$$\|d_V - d_{\tilde{V}}\|_\infty \leq 4\sqrt{k}\gamma\|A\|_{\infty,2}.$$

As $d_V \in \mathbb{R}^n$, we have $\|d_V - d_{\tilde{V}}\|_p \leq n^{1/p}\|d_V - d_{\tilde{V}}\|_\infty \leq 4\sqrt{k}\gamma n^{1/p}\|A\|_{\infty,2}$. Now, let

$$\mathcal{V}_S := \{\tilde{V} = \text{span}(\tilde{v}_1, \dots, \tilde{v}_k) \mid \tilde{v}_i \in N_S\}.$$

We have $|\mathcal{V}_S| \leq |N_S|^k \leq \exp(O(km\log 1/\gamma))$ since $|N_s| \leq \exp(O(m\log 1/\gamma))$. As there are $\binom{n}{m}$ choices for $S$, the total number of subspaces in the set $\cup_{S \in \binom{[n]}{m}} \mathcal{V}_S$ is upper bounded by $\exp(m\log n + km\log 1/\gamma)$. Using Lemma 4.1, using a union bound over all $\exp(m\log n + km\log 1/\gamma)$ choices of $\tilde{V}$, we have that with probability $\geq 99/100$, for all $\tilde{V} \in \cup_{\binom{[n]}{m}} V_S$,

$$\|\mathbf{D} \cdot d_{\tilde{V}}\|_\infty \geq \frac{\|d_{\tilde{V}}\|_p}{(\log 100 + m\log n + km\log 1/\gamma)^{1/p}}.$$

Using Lemma 4.1 again, we also have that $\max_i |\mathbf{D}_i| \leq C_3 n^{1/p}$ for a large enough constant $C_1$ with probability $\geq 99/100$. Condition on both these events. We have that for any $k$ dimensional subspace $V$ in the span of any set of $m$ rows of $A$,

$$\|\mathbf{D} \cdot d_V\|_\infty \geq \|\mathbf{D} \cdot d_{\tilde{V}}\|_\infty - \|\mathbf{D} \cdot (d_V - d_{\tilde{V}})\|_\infty$$
$$\geq \frac{\|d_{\tilde{V}}\|_p}{(\log 100 + m\log n + km\log 1/\gamma)^{1/p}} - C_1 n^{1/p}\|d_V - d_{\tilde{V}}\|_\infty$$
$$\geq \frac{\|d_V\|_p}{(\log 100 + m\log n + km\log 1/\gamma)^{1/p}} - \frac{4\sqrt{k}n^{1/p}\gamma\|A\|_{\infty,2}}{(\log 100 + m\log n + km\log 1/\gamma)^{1/p}}$$
$$- 4C_1 n^{1/p}\sqrt{k}\gamma\|A\|_{\infty,2}.$$

For any $Q$, we have that $\|d_V\|_p \geq \|d_V\|_2/\sqrt{n} \geq \|A - [A]_k\|_{\mathsf{F}}/\sqrt{n}$ using the fact that $Q$ is a $k$ dimensional subspace. Hence, if $\gamma \leq \text{poly}(\|A - [A]_k\|_{\mathsf{F}}/\|A\|_{\infty,2}, 1/n)$, then

$$\|\mathbf{D} \cdot d_V\|_\infty \geq \frac{\|d_V\|_p}{2(\log 100 + m\log n + km\log 1/\gamma)^{1/p}}.$$

Assuming $\text{rank}(A) \geq k$, we have that for any $n \times d$ matrix $A$ integer entries bounded by $\text{poly}(n)$, $\|A - [A]_k\|_{\mathsf{F}} \geq \text{poly}(n)^{-(k-1)/2}$. Hence, $\gamma$ can be taken as $1/\text{poly}(n)^k$ and we obtain that with probability $\geq 98/100$,

$$\|\mathbf{D} \cdot d_V\|_\infty \geq \frac{\|d_V\|_p}{2(\log 100 + m\log n + k^2 m\log n)^{1/p}}.$$

for all $k$-dimensional subspaces $V$ that lie in the span of $A_S$ for some $S \subseteq [n]$, $|S| \leq m = O(k^3 \log^2 n)$. $\qquad\square$

### B.3 WRAP-UP

Let

$$V^* = \underset{k\text{-dim subspaces } V}{\arg\min} \|d_V\|_p.$$

Condition on the event that $\|\mathbf{D}^{1/p}v_{P^*}\|_\infty \leq C_1\|v_{P^*}\|_p$ for a large enough constant $C_1$. The event holds with probability $\geq 99/100$ by Lemma 4.1. Finally, by a union bound, we have all the following events hold simultaneously with probability $\geq 9/10$:

1. Algorithm 1, when run on the rows of the matrix $\mathbf{D} \cdot A$, selects at most $m = O(k^3 \log^2 n)$ rows.

2. For any $k$ dimensional subspace $V$ contained in the span of any at most $m$ rows of $A$,

$$\|\mathbf{D} \cdot d_V\|_\infty \geq \frac{\|d_v\|_p}{C_2 k^{5/p} \log^{2/p} n}.$$

3. If $V^*$ is the optimal subspace that minimizes the $\ell_p$ norm of the distance vector to a $k$ dimensional subspace, then

$$\|\mathbf{D} \cdot d_{V^*}\|_\infty \leq C_1 \|d_{V^*}\|_p.$$

Conditioned on the above events, let $S \subseteq [n]$ be the coreset computed for the matrix $\mathbf{D} \cdot A$ by Algorithm 1. From Theorem 3.3, we have that for any rank $k$ projection matrix $P$,

$$\|(\mathbf{D}A)_S(I - P)\|_{\infty,2} \leq \|(\mathbf{D}A)(I - P)\|_{\infty,2} \leq Ck^{3/2} \log n \|(\mathbf{D}A)_S(I - P)\|_{\infty,2}.$$

Let $\hat{V}$ be the $k$ dimensional subspace defined as $\hat{V} \doteq \arg\min_{k\text{-dim } V} \|(\mathbf{D}A)_S(I - \mathbb{P}_V)\|_{\infty,2}$. Without loss of generality, we can assume that $\hat{V}$ is contained in the rowspace of $(\mathbf{D}^{1/p}A)_S$ and hence the row space of $A_S$. Therefore,

$$\begin{aligned}
\|A(I - \mathbb{P}_{\hat{V}})\|_{p,2} = \|d_{\hat{V}}\|_p \\
&\leq C_2 k^{5/p} \log^{2/p} n \|\mathbf{D} \cdot d_{\hat{V}}\|_\infty \\
&= C_2 k^{5/p} \log^{2/p} n \|(\mathbf{D} \cdot A)(I - \mathbb{P}_{\hat{V}})\|_{\infty,2} \\
&\leq C_2 \cdot C \cdot k^{5/p+3/2} \log^{1+2/p} n \|(\mathbf{D}A)_S(I - \mathbb{P}_{\hat{V}})\|_{\infty,2} \\
&\leq C_2 \cdot C \cdot k^{5/p+3/2} \log^{1+2/p} n \|(\mathbf{D}A)_S(I - \mathbb{P}_{V^*})\|_{\infty,2} \\
&= C_2 \cdot C \cdot k^{5/p+3/2} \log^{1+2/p} n \|\mathbf{D} \cdot d_{V^*}\|_{\infty,2} \\
&\leq C_1 \cdot C_2 \cdot C \cdot k^{3/2+5/p} \log^{1+2/p} n \|d_{V^*}\|_p.
\end{aligned}$$

Thus, $\hat{V}$ is an $O(k^{5/p+3/2} \log^{1+2/p} n)$ approximate solution for the $\ell_p$ low rank approximation problem over the matrix $A$.

---

**Algorithm 2:** Minimize the $\ell_p$ norm of the vector of distances to a $k$ dimensional subspace

---

**Input:** A matrix $A$ as a stream of rows $a_1, \ldots, a_n \in \mathbb{R}^d$, $p \geq 1$ and a rank parameter $k$
**Output:** Rank $k$ projection matrix $\hat{P}$
1 Feed the stream $\lceil e_1^{-1/p} \rceil a_1, \ldots, \lceil e_n^{-1/p} \rceil a_n$ to Algorithm 1 and obtain the set $S \subseteq [n]$
2 $\hat{V} \leftarrow \arg\min_{\substack{\text{rank-}k \\ \text{subspace } V}} \|(\mathbf{D} \cdot A)_S(I - \mathbb{P}_V)\|_{\infty,2}$
3 **return** $\hat{V}$

---

**Theorem B.1.** *Given a stream of rows $a_1, \ldots, a_n$, Algorithm 2 uses space necessary to store $O(k^3 \log^2 n)$ rows and outputs a rank $k$ subspace $\hat{V}$ satisfying*

$$\|A(I - \mathbb{P}_{\hat{V}})\|_{p,2} \leq C_1 \cdot C_2 \cdot C \cdot k^{3/2+5/p} \log^{1+2/p} n \min_{k\text{-dim } V} \|A(I - \mathbb{P}_V)\|_{p,2}.$$

## C  MISSING DETAILS ABOUT EXPERIMENTS

### C.1  MEASURING DISTORTION WITH IN THE SUBSPACE

Given a matrix $A$ and a parameter $k$, Algorithm 1 returns a coreset $S$. In our experiments we measure the maximum distortion defined as $\max_{x \in \text{rowspace}(A_S)} \|Ax\|_\infty / \|A_S x\|_\infty$. Since any vector in the rowspace of $A_S$ can be written as $A_S^\top y$ for some $y$, we want to measure $\max_y \|AA_S^\top y\|_\infty / \|A_S A_S^\top y\|_\infty$. Let the distortion be maximized at $y^*$ and that

$$\frac{\|AA_S^\top y^*\|_\infty}{\|A_S A_S^\top y^*\|_\infty} = \phi \geq 1.$$

Further let $i$ be the coordinate such that $\|AA_S^\top y^*\|_\infty = (AA_S^\top y^*)_i$. Now for each $j \in [n]$, consider the following linear program:

$$\min_{(y,t)} \quad t$$
$$\text{s.t. } a_j^\top A_S^\top y = 1$$
$$A_S A_S^\top y \leq t \cdot 1$$
$$-A_S A_S^\top y \leq t \cdot 1.$$

If $(y_j, t_j)$ is the optimum solution for the above problem, we note that $t_j = \|A_S A_S^\top y_j\|_\infty$. Since we have $a_j^\top A_S^\top y_j = 1$, we have that $\|AA_S^\top y_j\|_\infty \geq 1$ and therefore we have that $t_j = \|A_S A_S^\top y_j\|_\infty \geq \|AA_S^\top y_j\|_\infty / \phi \geq 1/\phi$. Thus for each $j \in [n]$, $1/t_j$ gives a lower bound on the maximum distortion $\phi$.

Now consider the linear program corresponding to $i \in [n]$ is defined above. Consider the vector $y = y^*/(AA_S^\top y^*)_i$. By definition, we have $a_i^\top A_S^\top y = a_i^\top A_S^\top y^*/(AA_S^\top y^*)_i = 1$ and $\|A_S A_S^\top y\|_\infty = \|A_S A_S^\top y^*\|_\infty / (AA_S^\top y^*)_i = \|A_S A_S^\top y^*\|_\infty / \|AA_S^\top y^*\|_\infty = 1/\phi$. Hence, $(y, 1/\phi)$ is a feasible solution for the linear program corresponding to index $i$. Since we proved above that $t_j \geq 1/\phi$ for all $j$, we get that $t_i = 1/\phi$ and hence $\max_j 1/t_j = \phi = \max_{x \in \text{rowspace}(A_S)} \|AA_S^\top y\|_\infty / \|A_S A_S^\top y\|_\infty$. In our experiments, we solve these linear programs and find the max-distortion within the rowspace of $A_S$.

## C.2 GRAYSCALE IMAGES USED

We use images from Leung (2017) and European Space Agency and NASA (2006) for our experiments. The compressed versions of the images used are in Figure 1. We include the full version of the images and our code which can be used to reproduce the results in the Supplementary material.

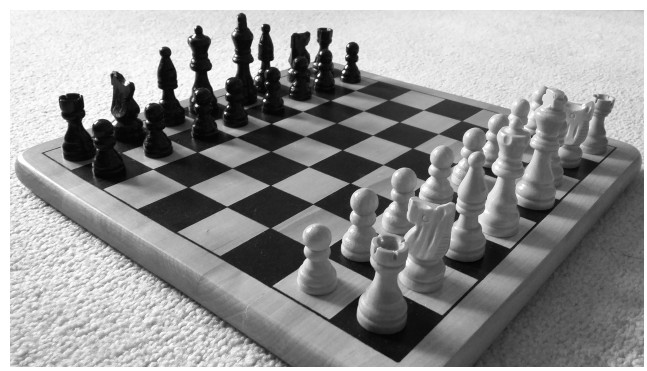

(a) Chessboard image from Leung (2017)

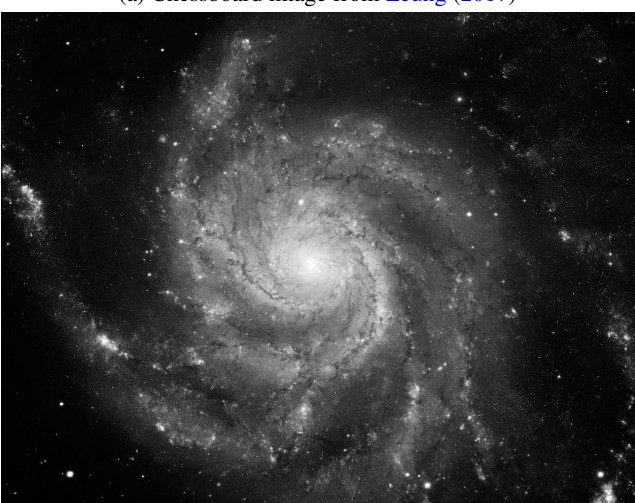

(b) Image of Pinwheel galaxy from European Space Agency and NASA (2006)

Figure 1: Images used for experiments

