# OpenReview forum: "High-Dimensional Geometric Streaming for Nearly Low Rank Data"
_ICLR.cc/2024/Conference — Submitted to ICLR 2024_

### Official Review · Reviewer_YvdX · 2023-10-31

**Soundness:** 4 excellent
**Presentation:** 3 good
**Contribution:** 4 excellent
**Rating:** 6
**Confidence:** 3

**Summary:**

This paper studies the outer (d - k)-radius estimation problem. This problem may be viewed as a clustering problem, where the goal is to find a k-flat F, such that the max distance of every data point to F is minimized. This is a generalization of the 1-center clustering, where k = 0. This paper gives a (strong) coreset for the problem with approximation ratio poly(k log n) and size poly(k logn). Here, A subset of points S is an \alpha-coreset, if for every k-flat F, the objective evaluated on S is in between [OPT, \alpha OPT]. This eventually implies a streaming algorithm that can achieve the same (order) of ratio.

Other extensions are also considered, and various similar results are obtained. Specifically, the main bound can be improved a little bit (in terms of the dependent in k and log n), by assuming a bounded rank-k condition number. Another extension is to consider an \ell_p aggregation function, which means the objective is changed to the sum of p-th power of distance of a data point to F. For this setting, a similar streaming bound can be obtained, but with a weaker error bound.

**Strengths:**

- The study is well-motivated. In particular, the problem is related to clustering and subspace approximation which are fundamental ML/data analysis tasks, and the streaming setting addresses the computational issues of ML in the big data era.

- The result can also be applied to improve a recent paper [17] in a certain case, which is a nice application that shows the theoretical relevance of the paper

- The paper also provides experiments, which indicate that the seemingly complicated steps can actually be implemented and have the potential to be used in practice

**Weaknesses:**

- The paper is quite technical and is not easy to understand especially for general audience. In addition, too many results are squeezed into the 9 pages. In my point of view, the author could focus on the main result Theorem 1.1, and this itself should already fit the volume of an ICLR paper (considering the 9 pages of the main text).

- I don't see a related work section. Since your main technique is coreset, it might make sense to mention works related to coreset.

- In fact, the discussion of the coreset literature is almost completely missing. Since your problem may be viewed as 1-center projective clustering, it's important to compare it with the relevant coreset literature. For example, this paper seems relevant: "New Coresets for Projective Clustering and Applications. Tukan et al. AISTATS 2022". From what I read, they gave O(1)-error coreset, but the size is k^k.

- It is not discussed if the coreset is tight or can be improved, in terms of the error bound

**Questions:**

- Does JL work here? In particular, can one reduce d to O(log n)? Or maybe subspace JL that reduce to O(k) dimension? I didn't find this discussed/mentioned. This is important as I guess otherwise your approach may not improve WS22? It may be useful to have a brief discussion of this in the paper.

- It seems Theorem 3.3 and Theorem 3.6 are in different models of streaming algorithms? Also, does the streaming algorithm in Theorem 3.3 work if deletions are allowed? Please clarify in the paper.

- In both Theorem 3.3 and Theorem 3.6 it is mentioned that the coordiantes are integers. Is this necessary even for the offline algorithm, or it is only a matter of storage model? Please clarify in the paper.

---

> ### Author Response · Authors · 2023-11-17
>
> - **Reorganizing the paper**: We will think about better ways to reorganize the paper to make it more understandable to the general audience. We think the applications to width estimation, convex hull approximation are important and should be present in the main paper and we will look for ways to make the presentation a bit less dense.
>
> - **Related work**: We will add a discussion about the existing coreset literature in the streaming/non-streaming settings for subspace approximation and related problems. We note that ours is the first strong coreset construction, with only poly(k, log n) points, for $\ell_{\infty}$ subspace approximation in the streaming setting and as we show the strong coreset property is important to obtain our applications.
>
> - **Projecting with JL**: It is not clear how to obtain strong coreset properties by projecting to lower dimensions using JL transforms in the streaming setting. Usually, projections with JL are used to obtain approximate sensitivities and then a strong coreset is constructed. But this cannot be implemented in the streaming setting with one pass. Kerber and Raghavendra (CCCG 2015) show that when we project to poly(k, log n) dimensions and construct a coreset, we can preserve the costs and extract a solution but we cannot approximate the cost of every subspace and obtain our application such as width estimation, convex hull etc.
>
> - **Streaming Model**: All the results in our paper are in the row-arrival model where we see one point after another. Our results do not hold when points can later be deleted.
>
> - **Integer Points**: It is not necessary for the points to have integer coordinates. We use bounded integer assumption to only obtain results that are independent of the condition number of the data. However, the algorithms will work even for arbitrary values and the size of the coreset constructed will depend on a specific condition number of the dataset.

---

> > ### Author Response · Authors · 2023-11-21
> > **Reminder**
> >
> > A reminder that the discussion period ends tomorrow. Please ask us if you have any questions about the paper and the rebuttal.

---

### Official Review · Reviewer_TTZL · 2023-11-02

**Soundness:** 4 excellent
**Presentation:** 4 excellent
**Contribution:** 3 good
**Rating:** 8
**Confidence:** 3

**Summary:**

The contributions of the paper are several-fold and it is indeed a bit hard to describe them succinctly... I will not go over all of them in this summary but give a basic flavor of what the paper does.

The paper considers several (somewhat) related problems in high-dimensional geometry and provides streaming algorithms for solving these problems. For some problems, like the "outer $(d-k)$-radius estimation problem", this paper provides the first streaming algorithms which use space that is poly$(k,d, \log (n))$ with a distortion factor also being poly$(k,d, \log (n))$. Here, we are given $n$ points in $\mathbb{R}^d$, along with a parameter $k < d$. Their algorithm works by constructing coresets using **online ridge leverage scores**.

For some other problems studied in the paper, prior work by Woodruff and Yasuda [FOCS 2022] already provided the first streaming algorithms using space poly$(d, \log (n))$ with a distortion factor also being poly$(d, \log (n))$. Woodruff and Yasuda also construct coresets but by using **online leverage scores**. In this submission, they show that assuming that the data points all approximately come from a low-rank subspace (which seems reasonable), their coreset algorithm can be used to provide streaming algorithms which are more efficient than the ones proposed by Woodruff and Yasuda.

**Strengths:**

Originality: First paper to provide streaming algorithms for the outer $(d-k)$-radius estimation problem. Their coreset algorithm is new and as they show has several applications. Would be of future interest to researchers working on other related problems.

Quality and Clarity: Paper is very well-written, easy to understand. I have not checked all the technical details in the proofs but they are very well-explained and it is unlikely that there are any major issues.

Significance: The paper's key contribution is providing a coreset construction algorithm, which they prove works for the problems which they consider, but also can be used independently (although without guarantees) for training other ML models.

**Weaknesses:**

Not any that I can see right now.

**Questions:**

None for now.

---

### Official Review · Reviewer_XBn6 · 2023-11-02

**Soundness:** 2 fair
**Presentation:** 2 fair
**Contribution:** 2 fair
**Rating:** 3
**Confidence:** 3

**Summary:**

In this paper, the authors consider the problem of designing a streaming algorithm for the $(d-k)$-dimensional radius approximation, i.e., find a $k$-dimensional flat that minimizes the maximum distance of any input point to this flat. When $k=0$, the problem is the well-known minimum enclosing ball problem, i.e., the smallest d-dimensional ball that encloses all input points. They also consider the $\ell_p$-sapce approximation problem. Overall, there are two major results that the authors claim:

1)	A single-pass streaming algorithm for $(d-k)$-radius estimation that has poly$(k,\log n)$ approximation using poly$(k, \log n)$ space. The algorithm maintains a coreset (a small subset of points) that approximates the $(d-k)$-radius in the streaming setting.
2)	The authors then extend this algorithm to the $\ell_p$-subspace approximation problem and design a streaming algorithm for this setting. The space requirement and the approximation ratios are poly$(k,\log n)$

**Strengths:**

The authors present a rather simple streaming algorithm for the $\ell_p$-subspace approximation problem and show its practical value via experiments.

**Weaknesses:**

The authors do not present the state-of-the-art for $\ell_\infty$-subspace approximation. For instance, when k is 0 or 1, there are O(1)-approximations (see for instance, Chan and Pathak CGTA 2014, Agarwal and Sharathkumar, (SODA 2010, Algorithmica, 2015) and some other followup work. Although for restricted k, they achieve significantly better approximation ratios. Does your algorithm achieve similar ratios when k is small? These algorithms are equally simple: Does your algorithm have a better empirical performance for instance for the MEB problem?

For width approximation, there are no randomized algorithm can achieve better than $d^{1/3}$-approximation while using $e^{d^{1/3}}$ space, this was shown in Agarwal and Sharathkumar, (SODA 2010, Algorithmica 2015). I think it is important to place your result in the context of this lower bound.

Overall, I had difficulty in understanding the impact and importance of this work and this, I believe, is mainly due to lack of a good comparison with existing work.

**Questions:**

Apart from the questions/concerns above, can you compare your work with the results in Kerber and Raghvendra (CCCG 2015):
https://research.cs.queensu.ca/cccg2015/CCCG15-papers/16.pdf
In particular, can one first apply JL-projection to d= O(poly{k, \log n}) dimensional space and then run a standard streaming algorithm for this space that gives O(d) =O(poly{k,\log n})-approximation in O(poly{k,\log n}) space?

---

> ### Author Response · Authors · 2023-11-17
>
> We will recap our main contributions here:
> - A **deterministic** coreset construction algorithm for $\ell_{\infty}$ subspace approximation with **any** rank parameter k. We are not aware of any previous works with streaming algorithms that work for any given rank parameter k.
> - A black-box **randomized reduction** from an $\ell_p$ subspace approximation coreset to an $\ell_{\infty}$ subspace approximation coreset for any rank k using scaled exponential random variables. This reduction is also new and may have additional applications since any improvement in an $\ell_{\infty}$ coreset construction would **directly improve** the $\ell_p$ coreset construction as well.
> - Deterministic Streaming Algorithms for other problems such as convex hull estimation, volume estimation, and width estimation in any direction, for almost low rank data with better guarantees than previous work of Woodruff and Yasuda (FOCS 2022)
>
> Our main motivation was to obtain streaming algorithms that work for values of k that are not just small constants as well. This particular problem seems to have not been studied in the streaming setting. We will add references to the line of work studying streaming algorithms for k=0,1 and add a discussion to place our results in context.
>
> - For k=0 and 1, we do not expect our algorithms to be better than the algorithms that are specifically constructed for those values of k. For MEB (k=0), our algorithm is essentially exactly the simple 2 approximation algorithm: pick an arbitrary point (in our case we pick the first) and essentially compute the distance of the farthest point from the first point. So, we do not expect to do better than the 2-approximation using our algorithm for k=0.
>
> - The width estimation problem, as studied in Agarwal and Sharathkumar (SODA 2014), corresponds to the case of k = d-1 for the $\ell_{\infty}$ subspace approximation problem. The hardness result of Agarwal and Sharathkumar is, as you stated, for approximating to a better factor than d^{1/3} in small space. Our streaming coreset construction gives only multiplicative approximation up to a factor of d^{3/2} in a bounded integer points case and a d^{1/2} factor for well-conditioned instances in this setting (k=d-1) and hence makes a poly(d) space streaming algorithm possible.
>
> **Comparison with Project-and-Construct Coreset**:
>
> For $\ell_{\infty}$ subspace approximation, ours is a **strong coreset** construction which can be used to approximate the subspace approximation cost of any **k-dimensional subspace**. As we show this strong coreset property has applications to width estimation in arbitrary directions, volume estimation etc.
>
> The random projection method in Kerber and Raghvendra (CCCG 2015)  only preserves the optimal projective clustering cost up to 1+eps approximation but cannot be used to approximate the projection cost of arbitrary subspaces.
> For $\ell_p$ subspace approximation though, our coreset construction can also not approximate costs for arbitrary inputs.

---

> > ### Author Response · Authors · 2023-11-21
> > **Reminder**
> >
> > A reminder that the discussion period ends tomorrow. Please ask us if you have any questions about our rebuttal.

---

### Meta-Review · Area_Chair_dtti · 2023-12-06

**Metareview:**

The paper proposes a new coreset algorithms for 1-center projective clustering with a wide range of parameters that can be applied to the streaming setting. In this problem, we are given a dataset in d dimensions and we would like to find a small subset (a coreset) such that for any given k-dimensional subspace, the maximum distance between the subspace and the subset is approximately the maximum distance between the subspace and the original data. The new algorithm also works in the streaming setting where we have low memory and can only scan through the dataset once. The paper also extends the result to the case where the clustering cost is an l_p norm of the vector whose entries are the distances as opposed to l_infinity (the maximum).

The reviewers all appreciate the importance of the particular clustering problem considered in the paper. They also note that the algorithm can be implemented and give reasonable results in the experiments.

On the downside, two reviewers note the lack of comparison with previous works even for special cases to show the significance of the improved guarantee. For example, when k=0 or 1, which are two common settings, the existing algorithms have much stronger guarantees. One reviewer also questions whether the bounds are close to tight, which is related to the previous point because the paper does not connect with previous lower bounds for the same problem.

**Justification For Why Not Higher Score:**

The lack of comparison with the existing literature is concerning as it seems there are a lot of previous works at least for the important special cases and the new work seems to give worse results for a more general setting and not a strict generalization. The positive reviewer suggests that the new algorithm is the first for the problem and can bring additional attention to the task but it seems that there is already significant related literature that was not thoroughly covered by the paper.

**Justification For Why Not Lower Score:**

N/A

---

### Decision · Program_Chairs · 2024-01-16

Reject